# COVID-19 Vaccination Program Data Analysis Based on Regional Status and Day Type: A Study from West Java Province, Indonesia

**DOI:** 10.3390/healthcare11050772

**Published:** 2023-03-06

**Authors:** Putri Adilla Ilhami, Mulya Nurmansyah Adisasmita, Dwi Agustian, Budi Sujatmiko

**Affiliations:** 1Undergraduate Medical Study Program, Faculty of Medicine, Universitas Padjadjaran, Sumedang 45363, Indonesia; 2Epidemiology and Biostatics Division, Department of Public Health, Faculty of Medicine, Universitas Padjadjaran, Bandung 40161, Indonesia

**Keywords:** COVID-19, vaccination program, regional status, day type

## Abstract

Vaccination is a strategy to control the COVID-19 pandemic and holds a crucial impact on global health. A better understanding of factors associated with vaccination is needed to establish a good vaccination program in a population. The purpose of this study is to analyze COVID-19 vaccination program data based on regional status and day type in the West Java Province of Indonesia and contribute to discovering other characteristics of the COVID-19 vaccination program. This study is a cross-sectional study using secondary data (N = 7922) from West Java’s COVID-19 Information and Coordination Center (PIKOBAR) from January to November 2021. Independent *t*-test with an alternative non-parametric Mann–Whitney U test (*p*-value < 0.05) is used as a statistical test in this study. The result reported significant differences in vaccination coverage between the city area and the regency area (*p* < 0.001). Significant differences in vaccination on working day and holiday were also found in both settings (*p* < 0.001). Vaccination was confirmed to be higher in the city compared to the regency and decreased on holiday compared to the working day. In conclusion, factors linked to regional status and day type must be considered as important factors for developing and accelerating vaccination programs.

## 1. Introduction

The global population has been enduring the devastating consequences of the Coronavirus Disease 2019 (COVID-19) pandemic since its first case detection in Wuhan, China [1,2]. Etiology of this disease is Severe Acute Respiratory Syndrome Coronavirus 2 (SARS-CoV-2), a type of coronavirus. This disease’s clinical manifestations vary from asymptomatic, mild to severe illness, with the most common mild symptoms being fever, sore throat, cough, tiredness, and anosmia or ageusia. Severe cases are more likely to develop in person with medical comorbidities as they may present pneumonia, hypoxemia, acute respiratory distress syndrome (ARDS), or multiorgan dysfunction [3,4,5,6,7,8]. This disease poses a serious threat to the healthcare system, as well as psychosocial, economic, and other aspects of life [9,10]. Indonesia, one of most populated countries in the world, is also greatly affected by COVID-19. The total number of COVID-19 cases globally as of November 2021 has surpassed 249 million cases, with a total of 4,249,323 cases reported across all provinces in Indonesia. The number continued to increase, reaching more than 649 million confirmed cases, and 6.6 million deaths had been reported globally as of 18 December 2022 [8,11,12].

Preventive measures have been taken in response to the rapid increase of reported cases, including health protocol implementation and vaccine development [13,14]. Vaccination is a promising strategy to control the pandemic and reduce COVID-19 morbidity and mortality rate [2,15]. COVID-19 vaccines are designed to form specific immunity against the SARS-CoV-2 virus for those who get vaccinated. The molecule contained in the vaccine acts as an antigen responsible for immune response stimulation [2,16]. Vaccination can also develop herd immunity if most of the population is vaccinated [14,17]. The target for COVID-19 vaccination coverage as of 2021 is 70% of the population according to World Health Organization (WHO) [8].

It is important to understand COVID-19 vaccination and its influencing factors to attain good vaccination coverage in the population. There are several factors related to vaccination, including knowledge, acceptance, hesitancy, geographical variation, healthcare capacity, and socioeconomic factors [10,18,19,20,21,22]. Some reports from the United States concluded that there are differences in vaccination coverage between urban and rural areas and differences in vaccination coverage between counties with different levels of social vulnerability [22,23,24]. In Indonesia, a study also reported the differences in basic immunization and measles vaccination between urban and rural areas [25,26]. These studies indicate that the characteristics of vaccine administration site should be considered since the difference in vaccination counts within an area may result from differences in location or population characteristics.

Vaccination number, as of 15 November 2021, had reached more than 7 million vaccine doses administered globally, with more than 215 thousand doses administered in Indonesia. The Indonesian government was targeting 234,666,020 citizens to get vaccinated, and as of 23 December 2022, the COVID-19 Task Force reported that 203,952,641 first dose vaccines, 174,666,157 s dose vaccines, and 68,201,141 first booster dose had been administered [11,12]. During those times, vaccination has been reported to undergo many changes, with the number of vaccine administrations varying daily and increasing cumulatively. Understanding the characteristics of day patterns may be helpful in addressing various factors affecting vaccination that may arise at various times.

The purpose of this study is to analyze vaccination number, vaccination coverage, and vaccination trend of COVID-19 in West Java based on regional status and day type for the period January–November 2021. An analysis of regional status will provide a better explanation of how location factors may be associated with vaccination, whereas analysis of day type will give additional knowledge about changes in vaccination patterns during a particular time or event. To the best of our knowledge, no research on COVID-19 vaccination for day type has been conducted in Indonesia. The result of this study hopefully will serve as additional data for evaluating the development of the COVID-19 vaccination program in West Java, Indonesia, and another area with similar circumstances.

## 2. Materials and Methods

### 2.1. Study Setting

Indonesia is one of the world’s most populous countries, with an area of 1.9 million km^2^ and a population of 270,203,917 as of 2020. It has 16,771 islands and 34 provinces divided into 416 regencies and 93 cities [27]. This study was conducted in the West Java Province of Indonesia, one of the provinces on Java Island. West Java provides adequate data to represent this study since it is a province with the largest population in Indonesia. This area consists of 27 regions, distributed into 9 cities and 18 regencies, with a population of 48,274,162 (17.87% of the total population) in 2020 [28]. Grouping of regional status serves as location variation observed in this study.

### 2.2. Study Design, Study Population and Data Collection

This is a cross-sectional study using secondary data from the Pusat Informasi dan Koordinasi COVID-19 Jawa Barat (PIKOBAR) as in West Java’s COVID-19 Information and Coordination Center, an e-government G2Z (Government to Citizens) application model created by West Java’s Digital Service [29,30]. The extracted data are the first and the second doses of COVID-19 vaccination data from 9 cities and 18 regencies in West Java. Overall, 7922 data points were collected in the form of daily vaccination count for each city and regency from 13 January 2021 to 23 November 2021.

### 2.3. Data Analysis

This study analyzed vaccination numbers, vaccination coverage, and vaccination trend in West Java. Vaccination number is a calculation of the total vaccines given to the population, vaccination coverage is the percentage of vaccination number per target population, and vaccination trend defines changes in vaccination count throughout time. The regional status and day type were taken into consideration when analyzing those variables. Regional status categorizes as city and regency, whereas day type is classified into two categories; working days (days from Monday through Saturday) and holidays (Sunday and national holidays determined by the government).

The data presented included the first dose and second dose vaccination, whereas the third dose or booster dose was not presented as its number was not significant enough and the vaccine had not been broadly distributed during the time of this study. Data were visualized using Tableau software Version 2022.1 and analyzed using R statistical software Version 4.2.1. Shapiro–Wilk normality test was initially done to assess normal data distribution and statistical analysis (*p*-value < 0.05) using independent *t*-test with an alternative non-parametric Mann–Whitney U test was chosen to find significant differences in vaccination based on regional status and day type.

## 3. Results

### 3.1. Vaccination Number and Vaccination Coverage

There are 7922 daily vaccination data from 9 cities and 18 regencies summarized in Table 1. Those data have been obtained since the first launch of public vaccination program [14]. A greater proportion of samples reside in the regency area (N = 5257) consisting of 18 regencies, whereas the city area (N = 2665) only has 9 cities registered.

The total calculation of vaccination number and vaccination coverage in West Java are depicted in Figure 1 and Figure 2. Those figures illustrate the vaccination count in West Java, showing the total number of vaccination and vaccination coverage in each region. Figure 1 shows the first dose vaccination count and Figure 2 shows the vaccination count with details of second-dose (completed dose) vaccination as follows: 1 city with a vaccination coverage of more than 80%, 4 cities and 1 regency with a vaccination coverage of 60–80%, 4 cities and 7 regencies with a vaccination coverage of 40–60%, and 10 regencies with a vaccination coverage of 20–40%.

Table 2 shows that there is no significant difference in vaccination numbers between the two regions for both first-dose vaccination (*p* = 0.160) and second-dose vaccination (*p* = 0.433). The differences in the median and interval range of vaccination count may be seen but not proven statistically significant.

Vaccination coverage data (%) are normally distributed (*p* > 0.05), therefore, the summarized data are presented as mean and standard deviation in Table 3. The city area has a higher mean than those in the regency area for both first-dose vaccination and second-dose vaccination. Following the statistical test, this study finds a significant difference in vaccination coverage (*p* < 0.001) between the two groups.

### 3.2. Vaccination Trend

Figure 3 depicts the trend of COVID-19 vaccination in West Java cities and regencies by first-dose vaccination numbers. Vaccination trend is counted since the first vaccination program began on 13 January 2021 and has increased over time.

An observable increase begins in March and significant decreases are found on Sundays and national holidays such as Lunar New Year, Prophet’s Ascension, Silent Day, and Good Friday. In April, a lower vaccination rate is seen at the beginning of Ramadan. Vaccination entering May is lower than the previous month, accompanied by decreases occurring on International Labor Day, Eid-Al Fitr 1442 H, and Vesak Day. Another increase is noticed at the end of June to July but continues to decrease on Sundays, Pancasila day, and Eid al-Adha 1442 H. Vaccination during the July–August transition tends to be stable, followed by an increase at the end of August and a decrease at the end of October. Vaccination trend for August–November period also show a decline on Sundays and national holidays such as the Islamic New Year 1443 H, Independence Day, and Prophet’s Birthday.

The statistical test results of vaccination based on day type are listed in Table 4 and Table 5. The normality test revealed that data are not normally distributed (*p*-value < 0.05), therefore, the non-parametric Mann–Whitney U test was used. Out of 7922 available data points, 6679 data were recorded during the working day type and 1243 data were recorded during the holiday day type (Table 4).

According to the data shown in Table 4, we can see differences in the median and maximal count of vaccination during working days and holidays. The statistical test found that those differences are statistically significant (*p* < 0.001).

Further observation was conducted to confirm differences in vaccination based on day type and regional status (Table 5). The numbers in Table 5 are the sum of daily vaccination data measured by accumulating the daily vaccination number within the same region and day. During the study period, 257 working days and 58 days of holiday, including Sunday, were identified.

The statistical test in Table 5 proves that there is a significant difference in vaccination during the working days and holidays (*p*-value < 0.001). It is also evident that the median and maximal count are lower on holidays than on the working days. These findings apply to both vaccination doses and occur in both settings.

## 4. Discussion

### 4.1. Analysis Based on Regional Status

West Java constitutes 18 regencies and 9 cities, and the government determines each regional status based on existing criteria [28]. Several aspects, including demographic, social, and economic, are considered when addressing an area as a regency or city. The city has a relatively larger population but a limited area with a higher population density than the regency. The majority of people in the city work in the trade or service sector, whereas the agricultural sector dominates in the regency [31]. The distinction between city and regency is analogous to the distinction between urban and rural areas, though not all regency areas are rural. This concept of regional disparity is still justifiable, regardless of the fact that West Java is generally considered an urbanized area.

The vaccination number of each city and regency in West Java (Figure 1 and Figure 2) is in line with the target population size of that region (Figure A1). It demonstrates that 9 of the 10 regions with a large population size have a higher first-dose vaccination number, namely Bogor Regency, Bandung City, Bekasi Regency, Bandung Regency, Bekasi City, Karawang Regency, Cianjur Regency, Garut Regency, and Sukabumi Regency. According to a study in Indonesia, one of the factors related to vaccination numbers is the size of the vaccination target, with a higher population target tending to have a higher vaccination number [32].

The result in Table 2 shows no significant difference in vaccination numbers, but Table 3 shows a significant difference in vaccination coverage between the city group and the regency group. It is essential to recognize that the vaccination number and vaccination coverage are parallel but different. The vaccination number is the total vaccine administered to the population that might be affected by population size, whereas vaccination coverage is the proportion of the vaccinated population. Vaccination coverage is preferred to represent vaccination achievement in a setting where different population size is identified.

The difference in vaccination coverage is shown to be statistically significant (*p* < 0.001) in Table 3. Another finding is also apparently seen in Figure 1 and Figure 2 where the city dominates the highest vaccination coverage, which may be related to the previously mentioned concept of regional disparity [33]. A similar study and report in the United States confirmed a difference in COVID-19 vaccination coverage between urban and rural areas associated with various factors such as social vulnerability, lower educational attainment, healthcare infrastructure, and access to healthcare facilities [22,23,24]. A study from Indonesia stated that the availability of healthcare facilities varying across different areas may lead to different vaccination rates [34]. Another study from China also discovered a significant difference in the level of access to healthcare among older adults in urban and rural areas [35]. The impact of healthcare capacity disparities was also mentioned as an important determinant of vaccine intake in a study from the United States [20]. Healthcare capacity is part of the health system, stated in the WHO “Framework for Action” on the health system. This framework explains six components of the health system, defined as Health System Building Blocks, which are service delivery, health workforce, health information, medical technologies, health financing, and leadership and governance. These building blocks serve as a convenient device for exploring and understanding the health system and the effects of interventions upon it, including such as on vaccination programs [36]. Referring to West Java’s statistics website, the number of healthcare facilities varies both in city and regency areas [28]. It could not yet be concluded whether the cities or regencies have a higher number of health facilities than each other, considering different size populations. A further study is needed to analyze the disparity of healthcare facilities between city and regency areas in West Java as well as the effect of health system capacity on the vaccination coverage in each area.

Some studies regarding COVID-19 vaccination furthermore mentioned that individual aspects such as knowledge, beliefs, acceptance, and hesitancy were associated with the rate of vaccine administration [10,18,21,37]. In Indonesia, because of frequent exposure to COVID-19 information, the urban community is known to have better knowledge about COVID-19 than those living in rural areas [38]. Several studies in Indonesia have confirmed a correlation between the level of knowledge and the public’s willingness to vaccinate. Another study also reported that the level of knowledge and confidence about COVID-19 was associated with vaccination adherence in Indonesia [39,40,41]. It could explain that a higher vaccination rate found in city areas of West Java may be due to a higher vaccine knowledge among the city population. Compared to other countries, a study in China also reported that urban employees with high vaccine knowledge are more likely to receive a vaccine booster shot [42]. A similar finding was also reported in Ethiopia, where the level of vaccine knowledge is significantly associated with vaccine acceptance among the adult population [43]. In addition, a study in Bangladesh revealed that people with good knowledge of vaccines showed higher odds of accepting vaccines than those with lower knowledge. This study also discovered lower odds of vaccine acceptance among respondents in a rural area in comparison with the urban area [44]. Along with the level of knowledge, a higher level of vaccine acceptance and positive attitude toward vaccines may also contribute to a higher vaccination rate in urbanized area. In conclusion, these findings imply that improving the level of public knowledge and attitude may be useful to raise awareness and the willingness to get vaccinated. Furthermore, prioritizing public health intervention in regencies or rural areas is needed to increase the rate of vaccine administration among the regency population.

### 4.2. Vaccination Trend Analysis and Its Relation to Day Type

The vaccination program in West Java underwent many changes during the period observed in Figure 3. As mentioned in the previous section, some studies stated that vaccine acceptance, intention, hesitancy, accessibility, availability, and healthcare capacity could contribute to changes in the vaccination rate [18,20,43,44,45]. Since the vaccination program started on 13 January 2021, vaccines are open and free for citizens, but some limitations still occurred [14]. A study in Indonesia reported that low vaccination coverage seen during January–February followed the limited vaccine availability in early 2021 [34]. Initial plan vaccine implementation at the beginning of the period was also restricted, allocated only for the prioritized population. Based on the Ministry of Health regulation, the vaccination program in Indonesia including West Java were arranged into three phases. The first phase that started from January was for health workers, assistants for health workers, and supporting personnel or medical students working in health care facilities, whereas the second phase started from the third week of February 2021 and was for the elderly and public service workers/officers. The third phase started from July 2021 and was for vulnerable people from geospatial, social, and economic perspectives, as well as for other communities [14]. Because of the schedules, the vaccination program in Indonesia showed a lower number during the early stage and started showing a significant increase after July 2021, when the program was open for public.

Moreover, this low initial vaccination rate may also be associated with a lack of knowledge and attitude toward COVID-19 vaccination, leading to vaccine hesitancy and rejection. According to several surveys conducted in Indonesia, the refusal by society caused the slow vaccination rate. Lack of information, anxiety about being the first to get vaccinated, and the worry surrounding unconfirmed side effects are several factors that lead to vaccine rejection [46]. The Indonesian Survey Institute conducted a survey from 20 to 25 June 2021 that reported 82.6% of Indonesian as not having been vaccinated, and the rest as not being willing to get vaccinated. The most cited reason was being afraid of the side effects, cited by 55.5% of respondents, and other common reasons were believing that the vaccine is not effective and thinking that they did not need it [47]. It was also similarly reported in France that a study found only 30.5% of their study population agreed to get vaccinated, whereas 31.1% declared being unsure of their vaccination intention during the first semester of 2021, proving that this also happened in another country [48].

In addition, a specific event, such as Ramadan, an Islamic fasting month, was learned to influence vaccination intention among the Muslim population [49]. Indonesians, a majority of whom are Muslims, took into consideration the halalness (permission) of vaccines. This happened in 2018 when there was a controversy regarding the halalness of the rubella vaccine. As for the COVID-19 vaccine, a survey reported that 48.39% of 9.39% of respondents who refused the vaccination doubted the halalness of the vaccine [50]. Although this could happen at other times than Ramadan, concern about this matter arose even more during Ramadan. The vaccination trend in Figure 3 slightly decreased when Ramadan began on 13 April 2021, and it remained low until May and the end of Ramadan. Concern about the safety and validity of the vaccine injection during fasting can lead to vaccine hesitancy, resulting in a decreased vaccination rate during Ramadan [49]. Concerning those matters, the Indonesian Ulema Council (Majelis Ulama Indonesia) declared the halalness of COVID-19 vaccines in Fatwa Number 2 of 2021 [50]. We have not been able to conclude the significant effect of Ramadan and religious authorities on vaccination rate in this study, but it is highly suggestive that considering the religious aspect may help improve the vaccination rate in religious countries. Given all the information presented above, a specified public health intervention at a particular time is necessary to counter any potential vaccine hesitancy or vaccine rejection issues that may arise.

In response to the low vaccination rate, the Indonesian government started implementing a vaccination center program to increase vaccination coverage in Indonesia. The government put lenient requirements to accelerate the vaccination program as of 24 June 2021. Additional vaccinators were also added on 11 September 2021, through the involvement of midwives and the National Population and Family Planning Board (BKKBN) to expand healthcare capacity [14]. West Java furthermore showed its contribution to accelerating the vaccination rate through a mass vaccination program called *Gebyar Vaksin* on 28 August 2021 [30]. Interestingly, several increases in vaccination rates coincided with the policies implemented at that time. A study conducted across 172 countries supports this observation by stating that the regulatory quality of a government is indeed a predictive power in explaining variations in vaccination rates across countries [51]. Furthermore, this connected with an insightful and applicable finding from a study of six countries that reported an association between mandatory COVID-19 certification and an increase in vaccination rate [52]. Although it was not stated in the figure presented in this study, several mandates had been already adopted in Indonesia at the time, including the implementation of mandatory vaccine certificates for traveling and entering the public area, such as department stores and public transportation [50]. The mandates were thought to be a promising strategy, as a study in the United States also reported that the major reason 35% of US adults got vaccinated was to participate in recreational activities that require vaccination proof, and 19% got vaccinated because of employer’s requirements [53]. This demonstrates how the government and stakeholders hold an impactful role in improving the vaccination program.

Based on day type, the vaccination trend illustrated in Figure 3 shows a constant decrease on every Sunday and national holiday. The statistical test shown in Table 4 confirms a significant difference between vaccination count on working days and holidays (*p*-value < 0.001). A further test was also performed and included in Table 5 using different data categorizations. After adding up vaccination count in the same day type based on its region, a significant difference in vaccination rate between working day and holiday day types both in city and regency areas were found. There has not been a similar study discussing this topic, but the level of vaccine service utilization may be able to explain why such a finding exists. It is known that during Sundays or holidays, most health workers do not work and most healthcare facilities do not open, resulting in a reduced availability of vaccination services. Other than that, according to the PIKOBAR website, almost all primary health facilities in West Java provide vaccination services only on a specific period between Monday and Saturday [30]. These conditions lead to lower accessibility and capacity of health services during Sundays and national holidays, thus altering the rate of vaccine administration during those times [20]. Despite the fact that people may get vaccinated elsewhere aside from primary health care, this study suggests that the likelihood of vaccine administration is reduced during the holiday day type.

Many healthcare interventions can be implemented with regards to several of the circumstances discussed above. Improving vaccine attitudes, increasing vaccine accessibility, and establishing an adaptive policy may become the options for enhancing vaccination programs. However, in addition to this, providing holiday access to vaccination is the most highlighted suggestion from this study. Increasing vaccination services is required to fix the low vaccination rate that constantly occurs during holiday days. A study from America about influenza vaccines aligned with this suggestion, as they found higher vaccination coverage in vaccination programs with weekend provision of vaccines and a train-the-trainer program [54]. This further correlates with a study from the United States that reported a higher number of patients younger than 65 years accessing vaccination services during off-clinic hours, including weekends and holidays [55]. A possible higher vaccine intention during holidays should be optimized by expanding access to vaccines and providing convenience for people in the population.

## 5. Conclusions

Several factors and events have influenced COVID-19 vaccination in West Java. Based on regional status, this study found a significant difference in vaccination coverage between the city and the regency area. In addition, observation of vaccination trends discovered changes in vaccination rates during a particular time, most highlighted on every Sunday and national holiday. The difference in vaccination rate during different day types is proven statistically, confirming a lower vaccination rate on holiday days compared to working days. Considering these findings, healthcare interventions, such as improving vaccine knowledge, acceptance, and willingness, should be prioritized in regencies or rural areas to increase vaccination coverage among the regency population. This study also advocates holiday access to vaccines to expand vaccine accessibility during holiday day type. Several example strategies are providing vaccine services in a 24-h-open public facility, a massive vaccination program on weekends such as *Gebyar Vaksin* mentioned before, or other special mandate-integrated programs supported by the government to increase vaccine accessibility during the holiday. Additionally, the discussion of vaccination trend analysis in this study encourages a comprehensive intervention in vaccination programs concerning public refusal issues, the effect of a particular event, and the regulatory capacity of the government. Those strategies can also be adjusted to increase the rate of booster vaccination that has been promoted nowadays.

## 6. Study Limitations

This study only provides data analysis proving the statistically significant difference in vaccination count. Due to limited data variables, demographic factors could not be analyzed, and the results can only be interpreted generally. It would be interesting to investigate the relationship between vaccination coverage and health system capacity in each area. We encourage further research to add demographic data or individual-based data to allow for a detailed interpretation. The spatial analysis also could be included in a further study analyzing the effect of health system capacity on vaccination coverage.

## Figures and Tables

**Figure 1 healthcare-11-00772-f001:**
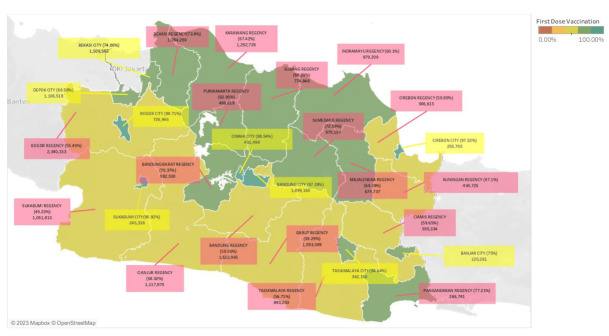
Total of first dose vaccination number and vaccination coverage in West Java as of 23 November 2021. The map color represents vaccination coverage and the box color represents regional status; pink for regency areas and yellow for city areas.

**Figure 2 healthcare-11-00772-f002:**
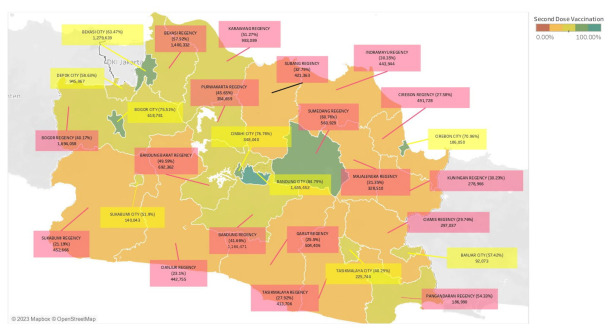
Total of second dose COVID-19 vaccination number and vaccination coverage in West Java as of 23 November 2021. The map color represents vaccination coverage and the box color represents regional status; pink for regency areas and yellow for city areas. A statistical test was carried out to analyze the significant difference in vaccination numbers and vaccination coverage between the city and the regency. The result of the Saphiro–Wilk normality test has shown that the vaccination number data in the two regions are not normally distributed (*p* < 0.05), whereas vaccination coverage data are normally distributed (*p* > 0.05). The statistical test used in Table 2 and Table 3 is adjusted to the result of the normality test.

**Figure 3 healthcare-11-00772-f003:**
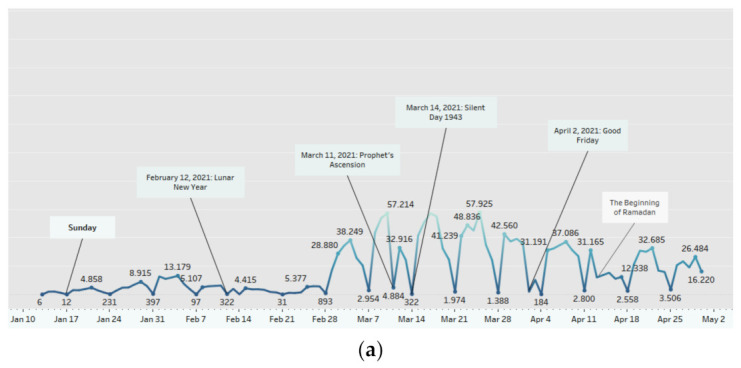
First-dose vaccination trend in West Java: (**a**) January–April 2021; (**b**) May–July 2021; and (**c**) August–November 2021. The marks are labeled by the sum of first-dose vaccination count; the dates shown are labeled by Sunday.

**Table 1 healthcare-11-00772-t001:** Summary of vaccination data in West Java (13 January–23 November 2021).

	N = 7922	Min–Max	Median	Mean	St. Dev
**Vaccination Number**					
**City**					
First Dose	2665	0–98,831	902	2487	4828
Second Dose		0–81,210	597	2061	4121
**Regency**					
First Dose	5257	0–62,804	1269	3389	5774
Second Dose		0–47,591	620	2114	4036
**Vaccination Coverage (%)**					
**City**					
First Dose	2665	0–97.5	17.2	30.9	0.3
Second Dose		0–84.8	11.1	19.8	0.2
**Regency**					
First Dose	5257	0–77.2	7.7	16.6	0.2
Second Dose		0–60.8	3.7	8.5	0.1

N = count of daily vaccination data recorded in each city and regency; vaccination number = daily single count; vaccination coverage = daily cumulative count.

**Table 2 healthcare-11-00772-t002:** Statistical analysis of total vaccination number based on regional status.

	N = 27	Median	Min–Max	*p* Value **
**First Dose**				
City	9	401,490	120,216–1,899,180	0.160
Regency	18	892,912	265,741–2,340,153	
**Second Dose**				
City	9	348,040	92,073–1,655,452	0.433
Regency	18	617,332	186,990–1,694,058	

N = count of region; *p* < 0.05; Confidence Interval = 95%; ** = Mann–Whitney U test.

**Table 3 healthcare-11-00772-t003:** Statistical analysis of total vaccination coverage based on regional status (%).

	N = 27	Mean	St. Dev	*p* Value *
**First Dose**				
City	9	82.9	12.3	<0.001
Regency	18	61.1	8.5	
**Second Dose**				
City	9	64.4	13.9	<0.001
Regency	18	37.8	12.6	

N = count of region; *p* < 0.05; Confidence Interval = 95%; * = Independent *t*-test.

**Table 4 healthcare-11-00772-t004:** Statistical analysis of vaccination data based on day type.

Characteristics	N = 7922	Median	Min–Max	*p* Value **
**First Dose** Working Days Holidays				
	6679	1429	0–98,831	<0.001
	1243	45	0–48,471	
**Second Dose** Working Days Holidays				
	6679	830	0–81,210	<0.001
	1243	12	0–39,664	

N = count for daily vaccination data recorded in each city and regency; Working days = Monday-Saturday; and Holidays = Sunday and national holidays. *p* < 0.05; Confidence Interval = 95%; ** = Mann–Whitney U test.

**Table 5 healthcare-11-00772-t005:** Statistical analysis of total daily vaccination based on day type in both city and regency settings.

Characteristics	N = 315	Median	Min–Max	*p* Value **
**First Dose**				
**City**				
Working Days	257	16,403	3–177,573	<0.001
Holidays	58	1565	1–63,877	
**Regency**				
Working Days	257	37,601	3–296,315	<0.001
Holidays	58	2027	0–66,950	
**Second Dose**				
**City**				
Working Days	257	13,370	0–126,852	<0.001
Holidays	58	470	0–66,950	
**Regency**				
Working Days	257	16,384	0–227,206	<0.001
Holidays	58	1243	0–49,283	

N = count of day; Working days = Monday-Saturday; Holidays = Sunday and national holidays; *p* < 0.05; Confidence Interval = 95%; ** = Mann–Whitney U test.

## Data Availability

All the data involved in this paper were obtained through cooperation with West Java’s Digital Services. If readers need more information about data and materials, please visit https://pikobar.jabarprov.go.id/ (accessed on 2 January 2022) or contact author for reasonable data requests.

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
