# Peer review of "COVID-19 Vaccination Program Data Analysis Based on Regional Status and Day Type: A Study from West Java Province, Indonesia"

_healthcare, 2023, doi:10.3390/healthcare11050772_

Round 1

Reviewer 1 Report

The authors submitted the report of an observational study conducted on the extent of vaccination in West Java, Indonesia.

The manuscript is well written. I have some comments for the authors in order to improve the quality of the manuscript:

1. Please provide more information on vaccines availability and vaccination strategy adopted in West Java. Were vaccines free of charge for everyone? Did the government offered the vaccines to different categories of population during different period? In some countries, vaccination was not allowed contemporarily to the whole population, and this is a main determinant of timing distribution of vaccination.

2. Please clarify whether you find a significant difference between cities and regencies. From Table 2 it seems not statistically significant, but the text says the contrary in many places (e.g. discussion). Is there any mistake? Can the authors clarify this result?

3. Table 2 is not clear. I suggest labelling the raw as first dose and second dose inside the table instead of clarifying this in the footnotes.

4. I totally agree with the limitations. It could have been interesting to note age and gender distribution also, or socio-economical issues. Many variables that the authors declared not to have access to, may be confounders in this analysis. 

5. Please note that some figure 1 is difficult to read due to graphical quality. The other figures are ok.

Reviewer 2 Report

Estimated Authors,

I've read with great interest your study entitled "COVID-19 Vaccination Program Data Analysis based on Regional Status and Day Type: A Study from West Java Province, Indonesia".

Authors have performed an accurate description of the vaccination trends in a highly populated Area of Southern Asia, identifying an interesting time trend in vaccination rates.

Still, the paper is affected by some substantial shortcomings that impair its acceptance in the current stage of development. 

More precisely:

Authors have performed an extensive description of the vaccination time trend focusing on the effect of national and religious holidays on vaccination rates, while more limited analyses are performed on what would be of particular interest, i.e. the effect of interventions performed in order to improve vaccination rates. According to the discussion, rows 262-272, several interventions in accord with religious authorities were performed: analyses should be performed in order to compare the effect of these intervention on the background vaccination rates. That would radically improve the significance for international readers (i.e. "if we involve religious authorities, can we improve the efficacy of vaccination campaigns?"). Moreover, several further information are needed to the international readers to understand the significance of this study: how was the SARS-CoV-2 vaccine delivered? which vaccines were performed (i.e. mRNA, adenoviral, inactivated ones?)? Were fake news or did specific episodes occur during the vaccination campaign? For instance, in Europe vaccination campaign was extensively compromised by the side effects allegedly associated with Astra Zeneca vaccines, and the acceptance of the formulates did substantially slow down for severa weeks.

Moreover:

- the paper (at the moment), only reflects the status of vaccination campaign at November 2021; even though the reported figures stop at the end of 2021 (no problem about it) introduction and discussion should report the status of the SARS-CoV-2 pandemic at least by december 2022, and similarly also the status of vaccination campaign in Indonesia by beginning of 2023;

- histograms included by Authors are quite clear, but as international readers may be quite unfamiliar with the areas, some maps including vaccination rates could radically improve the readability of the paper;

- when discussing the vaccination rates in urban vs. rural areas, Authors should provide information about the infrastructures that were involved, as it would be quite expected that areas more difficult to reach for getting vaccinated had lower rates, and conversely.

Round 2

Reviewer 1 Report

The authors addressed all my previous comments. I have no further concerns.

Reviewer 2 Report

Authors have mostly addressed my previous concerns.

I'm still quite doubtful about the eventual aims of this paper (that in its current status is more consistent with the definition of real-world report than with that of a research study), but the overall quality is quite proper for endorsing its eventual acceptance.